# AudioStory: Generating Long-Form Narrative Audio with Large Language Models

## Abstract

Recent advances in text-to-audio (TTA) generation excel at synthesizing short audio clips but struggle with long-form narrative audio, which requires temporal coherence and compositional reasoning. To fill this gap, we propose AudioStory, a unified framework that integrates large language models (LLMs) with TTA systems to generate structured, long-form audio narratives. AudioStory possesses strong instruction-following reasoning generation capabilities. It employs LLMs to decompose complex narrative queries into temporally ordered sub-tasks with contextual cues, enabling coherent scene transitions and emotional tone consistency. AudioStory has two appealing features: (1) Decoupled bridging mechanism: AudioStory disentangles LLM-diffuser collaboration into two specialized components, *i.e.*, a bridging query for intra-event semantic alignment and a residual query for inter-event coherence preservation. (2) End-to-end training: By unifying instruction comprehension and audio generation within a single end-to-end framework, AudioStory eliminates the need for modular training pipelines while enhancing synergy between components. Furthermore, we establish a benchmark AudioStory-10K, encompassing diverse domains such as animated soundscapes and natural sound narratives. Extensive experiments show the superiority of AudioStory on both single and narrative audio generation, in terms of instruction-following ability and audio fidelity. Our code and dataset will be publicly available.

## 1 Introduction

Audio content plays a pivotal role in modern media, from immersive storytelling and podcasts to interactive entertainment and education. Recent advancements in text-to-audio (TTA) generation, exemplified by models such as TangoFlux (Hung et al., 2024), AudioLDM (Liu et al., 2024), and Stable Audio (Evans et al., 2024), have demonstrated remarkable capabilities in synthesizing high-quality, short-form audio clips from textual descriptions. However, a critical gap remains in generating long-form narrative audio, *i.e.*, coherent, structured sequences of audio instances that unfold over extended durations, such as audiobooks, podcasts, or dynamic soundscapes for games.

Long-form narrative audio generation introduces unique challenges that extend beyond single-prompt synthesis. First, it requires temporal coherence: maintaining consistency in themes, sound effects, and emotional tone across the whole audio. Second, it demands narrative reasoning to decompose a complex instruction into logically ordered sub-events, characters, or environmental interactions. For instance, a prompt like "A suspenseful chase through a rainstorm: footsteps splash, thunder roars, a car skids, and a door slams shut" necessitates not only generating individual sounds but also orchestrating their timing, intensity, and interplay to build tension. Existing TTA models, while proficient at capturing isolated events, often struggle with such compositional and temporal reasoning, leading to fragmented or inconsistent outputs.

To address these challenges, we propose AudioStory, a novel multi-step framework for generating long-form narrative audio by integrating the reasoning capabilities of LLMs with audio generation. As shown in Fig. 1, we propose *interleaved reasoning generation* following a divide-and-conquer manner: reasoning for general narrative plans, decomposing plans into sequential generation actions, and generating interleaved audio events step-by-step. Specifically, AudioStory employs LLMs to decompose a narrative query (in language or multimodality) into a structured sequence of audio-generative sub-tasks, each accompanied by contextual cues such as temporal offsets, emotional tone,

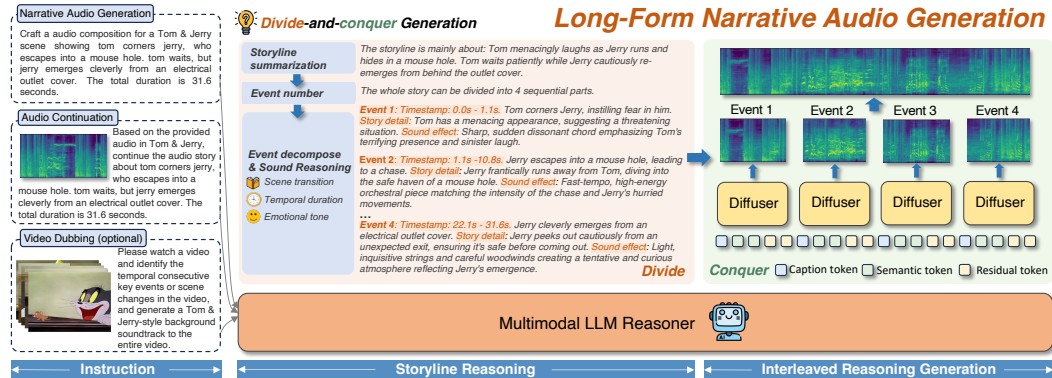

Figure 1: AudioStory effectively follows multimodal instructions, decomposing them into a sequence of coherent audio segments that capture scene transitions, affective tone, and precise timestamps. Unlike prior T5-based diffusion models that falter on complex queries, AudioStory endows LLMs with explicit high-level planning, enabling robust instruction-following and temporally consistent long-form audio generation. Video dubbing constitutes an extended application of the framework.

and character interactions. These reasoning chains are then synthesized into audio events using a diffusion backbone, with explicit mechanisms to ensure style consistency, smooth transitions and temporal alignment. We streamline the narrative planning via LLMs and audio synthesis via diffusion models into an end-to-end framework, enabling the generation of rich, multi-scene audio stories that adhere to user intent while preserving coherence over time.

AudioStory introduces several technical innovations: First, unlike prior arts (Wu et al., 2024; Lai et al., 2024) that bridge LLMs with audio diffusers through predefined textual spaces (Raffel et al., 2023), we propose a decoupled bridging space consisting of two distinct tokens: (1) *semantic tokens*, which encode text-oriented audio semantics, and (2) *residual tokens*, which capture nuanced acoustic cues and cross-event correlations. This design effectively improves both audio fidelity and temporal consistency during generation. Second, unlike zero-shot integration of LLMs and diffusers, our framework supports end-to-end progressive training, enabling joint optimization of instruction understanding and audio synthesis. This synergistic training paradigm enhances both audio understanding and generation. Third, we introduce the first narrative audio generation benchmark, providing a comprehensive evaluation for assessing audio generation quality and consistency.

The contributions of the paper are as follows:

- We introduce AudioStory for narrative audio generation, which integrates LLM-based reasoning and iterative diffusion-based generation in a unified framework, with strong multimodal instruction-following and audio generation abilities.

- We propose decoupled bridging tokens for LLM-diffuser collaboration, using semantic tokens (text-oriented audio semantics) and residual tokens (nuanced acoustic cues) to improve audio fidelity and temporal consistency.

- We introduce a synergistic training paradigm, facilitating collaboration and complementarity between LLM and diffusion models. Unlike zero-shot LLM-diffusion integration, our framework enables end-to-end joint training, enhancing both multimodal understanding and generation.

- Experiments show AudoStory significantly surpasses prior diffusion-based and MLLM-based models by a large margin in narrative audio generation. We also uncover some important findings across multiple aspects, including reasoning formulation, bridging mechanism and training recipes.

## 2 RELATED WORKS

**Text-to-audio generation (TTA).** Recent advances in generative models have significantly advanced text-to-audio generation. Make-An-Audio (Huang et al., 2023) and AudioLDM (Liu et al., 2023; 2024), synthesize audio through iterative denoising of text-conditioned latent representations. Tango (Majumder et al., 2024; Ghosal et al., 2023), Audio Flamingo (Kong et al., 2024), GenAu (Haji-Ali et al., 2024), Fugatto (Valle et al., 2025) further enhance design spaces of latent space, data quality

and cross-modal alignments. Recently, Stable Audio series (Evans et al., 2024) employs hierarchical latent diffusion trained on large-scale datasets for high-fidelity output. Beyond diffusion-based priors, flow-matching techniques optimize probability density transport for audio synthesis. VoiceBox (Le et al., 2023) enables zero-shot style transfer via continuous normalizing flows. TangoFlux (Hung et al., 2024) introduces CLAP-ranked preference optimization to enhance text-audio alignment. Existing methods align text and audio semantically but primarily target descriptive queries, limiting interactive control and adaptability to evolving instructions. They are also confined to short audio domains. These limitations demand TTA models to handle complex instructions over long durations.

**Any-to-any multimodal LLMs.** *Any-to-any* models (Tang et al., 2023a; Wu et al., 2024; Zhan et al., 2024; Lai et al., 2024; Ge et al., 2023) aim to accept arbitrary input modalities and generate outputs in any desired modality. Pioneering efforts include CoDi (Tang et al., 2023b;a) leveraged composable diffusion for diverse modality handling. Spider (Lai et al., 2024) further enables the generation of multiple modalities in a single response. NExT-GPT (Wu et al., 2024) demonstrated the efficacy of lightweight alignment for adapting LLMs to multimodal tasks, while AnyGPT (Zhan et al., 2024) showcased the potential of discrete sequence modeling. Unified-IO2 (Lu et al., 2023) highlighted the impact of scale and unified architectures in achieving remarkable performance across many tasks. Despite these advancements, current methods exhibit limitations in long-context generation with complex instructions: First, they primarily focus on speech generation and simple caption-to-music or caption-to-sound tasks, struggling to comprehend general and intricate human instructions beyond basic captions. Second, their audio generation is typically limited to single, short segments, hindering the generation of longer audio sequences.

# 3 NARRATIVE AUDIO GENERATION

**Problem definition.** Narrative audio generation aims to generate long-form, structured and temporally coherent audio sequences $A = \{A_m\}_{m=1}^{M}$, given multimodal instruction $x_{\text{ins}}$ (*e.g.*, language, audio or vision), where $M$ is the number of audio segments. The task shares a similar formulation with the text-to-audio generation, but is far more challenging due to two distinct capabilities: (1) Temporal coherence, *i.e.*, maintaining consistency in themes, sound effects, and emotional tone across extended durations; (2) Compositional reasoning. *i.e.*, decomposing high-level narrative instructions into logically ordered events (*e.g.*, "footsteps splash, then thunder roars") with precise timing and contextual interactions. Existing TTA systems, while effective for short clips, lack explicit mechanisms to model cross-segment dependencies or align audio events with evolving narrative structures, limiting their applicability to real-world scenarios.

**The AS-10k benchmark.** Given the lack of quantitative evaluation, we establish the AS-10k benchmark for the narrative audio generation task. AS-10k comprises 10k annotated audios paired with narrative prompts. We collect videos from two primary sources: (1) **Natural sounds**: We select 4,723 audio instances from UnAV-100 (Geng et al., 2023), covering a broad spectrum of real-world environmental recordings (*e.g.*, rainstorms, animal calls, rustling leaves) and human activities (*e.g.*, footsteps, door slams, and conversations). This collection ensures sufficient coverage of everyday acoustic events and ambient soundscapes. (2) **Animated sounds**: We curate 5,332 audio clips from 157 episodes of *Tom & Jerry*, capturing stylized background music (*e.g.*, orchestral pieces, string sections) and sound effects (*e.g.*, slapstick actions, cartoonish collisions and rapid movements). These animated sounds feature stylized and expressive audio content, distinct from natural sound recordings.

The annotation pipeline involves three stages. First, we filter the videos with sequential audio events, ensuring the storyline of the audio is visually-grounded for meaningful activities. Then, we parse the video into several key audio events by Gemini-2.5-Pro (Team et al., 2023), each is labeled with its timestamps, audio caption and visual captions. Next, given these text-based timestamped captions, we prompt GPT-4o (OpenAI, 2025) to generate diverse instructions and chain-like reasoning steps.

To be specific, we design diverse formats of multimodal instructions, including text-only instructions for narrative audio generation, audio-text ones for audio continuation and video-text ones for video dubbing, as in Fig. 1. For a flexible control of duration and semantic elements of generated audios, we make the intermediate reasoning encompass at least the following steps: *storyline summarization* for global summarization of general story, *event decomposition* for inferring the number of audio events, *sound reasoning* for predicting timestamp and key elements (*e.g.*, emotional tone, scene transition) of each event. All detailed prompts and processing steps are in Appendix H.1.

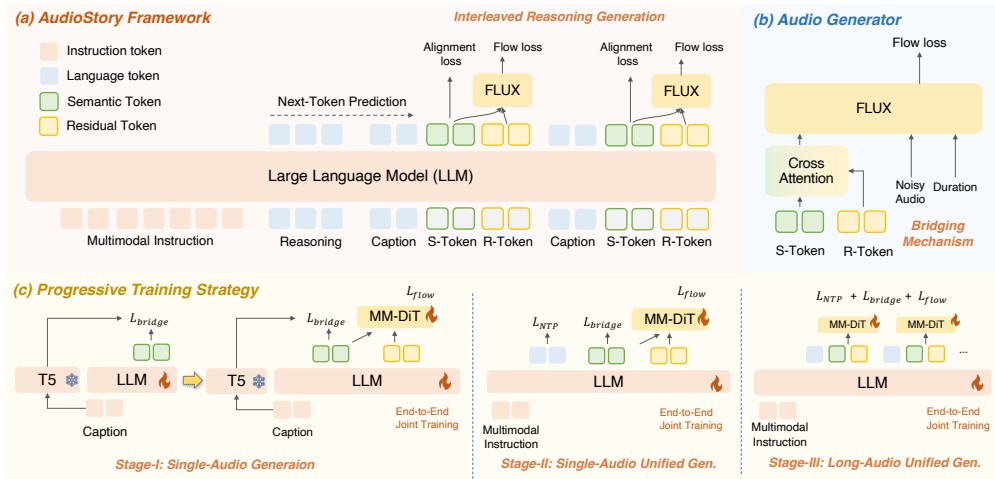

Figure 2: Overview of AudioStory with three core components: (a) The LLM processes the instruction input, decomposes the long audio into structured sub-tasks, and sequentially generates a caption, semantic tokens, and residual tokens for each audio clip. (b) After fusing semantic and residual tokens, they are combined with the duration information as conditioning inputs to the DiT, which then generates each audio clip. (c) The progressive training recipe with three stages.

**Evaluation metrics.** The AS-10k dataset includes 5.3k samples of natural sounds and 4.7k samples of cartoon audios. We randomly divided the dataset into 85% for training and 15% for testing. We devise a comprehensive evaluation spanning three aspects: *instruction-following*, *consistency*, and *generation quality*. We employ Gemini-2.0-flash as the evaluator with a score range of 0-5 for these metrics. More details could be found in the Appendix H.2.

## 4 AUDIOSTORY

**Overview.** To achieve instruction-followed audio generation, the ability to understand the input instruction and reason about relevant audio sub-events is essential. To this end, AudioStory adopts a unified understanding-generation framework (Fig. 2). Specifically, given multimodal instructions, an LLM analyzes and decomposes it into structured audio sub-events with context. Based on the inferred sub-events, the LLM first performs interleaved reasoning generation (Sec. 4.1), sequentially producing captions and bridging tokens between the LLM and the audio generator (Sec. 4.2). Through progressive end-to-end training, AudioStory ultimately achieves both strong instruction comprehension and high-quality audio generation (Sec. 4.3).

### 4.1 INTERLEAVED REASONING GENERATION

Directly generating long-form narrative audio that aligns with complex instructions is challenging. We take the spirit of "divide-and-conquer" and propose decoupling the input instruction into chronological short audio clips, which are then combined to form the complete long-form narrative audio.

**Single-audio clip generation.** The ability to generate individual audio clips from captions is a foundational step toward producing sequential audio events. For audio clip generation, the LLM generates bridge tokens from a given caption, which serve as conditions for the DiT. While this method works well for short audio generation based on simple captions, it becomes insufficient for complex instructions involving multiple events, temporal relationships, or narrative structures.

**Interleaved reasoning generation for long-audio generation.** We propose to decouple a complex, long-form audio into multiple audio segments for segment-by-segment generation. This divide-and-conquer process consists of two components: (1) *Storyline reasoning*: LLMs reason through the entire instruction, inferring the number of audio events. Furthermore, LLMs analyze the start and end timestamps of each event, as well as the event description and corresponding audio content that should be included. (2) *Interleaved generation*: For each event, the LLM infers the caption, duration,

and corresponding bridge queries (semantic tokens and residual tokens, as described in Sec. 4.2), enabling interleaved generation. These queries, along with duration information, are then provided as conditional inputs to the DiT-based audio generator. By accurately predicting durations and utilizing semantically rich bridging tokens, the model ensures both coherent audio semantics within each event and consistency across events. The training data is structured as:

$$[\texttt{BOS}][\texttt{BOT}]\{\#\text{event}\}\{\text{storyline reasoning tokens}\}[\texttt{EOT}][\texttt{BOG}]\{\text{caption}\}\{\text{duration}\}$$
$$\mathbf{T}_{\text{semantic}}\mathbf{T}_{\text{residual}}[\texttt{EOG}]\cdots[\texttt{BOG}]\{\text{caption}\}\{\text{duration}\}\mathbf{T}_{\text{semantic}}\mathbf{T}_{\text{residual}}[\texttt{EOG}][\texttt{EOS}]. \tag{1}$$

The textual tokens in the entire reasoning process is supervised by the next token prediction loss:

$$\mathcal{L}_{\text{reason}} = \mathcal{L}_{\text{text}}^{\#\text{event}} + \mathcal{L}_{\text{text}}^{\text{content}} + \mathcal{L}_{\text{text}}^{\text{caption}}, \quad \text{where} \quad \mathcal{L}_{\text{text}} = \prod_{i=1}^{L} p(\boldsymbol{x}_i | \mathbf{X}_{<i}, \mathbf{X}_{p,<i}). \tag{2}$$

## 4.2 DECOUPLED BRIDGING MECHANISM

Once the LLM is capable of effective reasoning, establishing a seamless bridge between the LLM and the DiT becomes crucial. However, text *alone* might not be the optimal bridge. Although it carries rich semantics, it fails to capture diverse low-level details of the audio modality, *e.g.*, timbre, rhythm, and ambience. Consequently, we propose decoupled bridges queries, which could be divided into semantic $\mathbf{T}_{\text{semantic}}$ and residual tokens $\mathbf{T}_{\text{residual}}$. The semantic tokens represent the audio's high-level semantics, while the residual tokens carry low-level audio details. They complement each other, enabling the disentanglement of audio information. In practice, after producing the caption for each audio event, the LLM collectively generates semantic and residual tokens. For semantic tokens, we use the textual tokens from Flan-T5 (Raffel et al., 2020) $\mathbf{T}_{\text{semantic}}^{\text{gt}}$ as the supervision using MSE loss:

$$\mathcal{L}_{\text{mse}} = \|\mathbf{T}_{\text{semantic}}^{\text{gt}} - \mathbf{T}_{\text{semantic}}\|_2^2. \tag{3}$$

The residual tokens are employed to supplement the missing information of the semantic tokens. Then, both types of tokens are merged and fed into as the conditional inputs of DiT. Here, we adopt multi-head cross-attention to merge these two tokens and obtain the resultant bridge queries:

$$\mathbf{H}_{\text{bridge}} = \texttt{Cross-Attn}(\mathbf{T}_{\text{semantic}}, \mathbf{T}_{\text{residual}}, \mathbf{T}_{\text{residual}}). \tag{4}$$

For audio generator with $\mathbf{H}_{\text{bridge}}$ as condition, we employ flow-matching (Esser et al., 2024) for generative modeling:

$$\mathcal{L}_{\text{flow}} = \mathbb{E}_{\boldsymbol{x}_1, \boldsymbol{x}_0, t} \|u(\boldsymbol{x}_t, t, \boldsymbol{c}) - \boldsymbol{v}_t\|_2^2, \tag{5}$$

where $\boldsymbol{c}$ is the condition and we choose $\boldsymbol{c} = \mathbf{H}_{\text{bridge}}$ and $t$ is uniformly sampled from $[0, 1]$. Through the generative supervision, $\mathbf{T}_{\text{residual}}$ can capture detailed information and complement $\mathbf{T}_{\text{semantic}}$.

## 4.3 PROGRESSIVE TRAINING STRATEGY

After establishing an effective bridge between the LLM and DiT, it becomes essential to design an efficient end-to-end training mechanism to build synergy between the understanding and generation tasks. We propose a progressive training strategy, following a single-to-multi and generation-to-unification paradigm. The training could be divided into three stages, where the model (1) learn to generate single audio segments, followed by (2) unified generation and understanding for single audios and (3) long-audio adaptation.

**Stage-I: Single-audio generation.** There are two sub-stages. (1) Stage-I-Warm, AudioStory learns to generate semantic tokens with MSE supervision in equation 3. Only the LoRA of the LLM and the projector of $\mathbf{T}_{\text{semantic}}$ are updated. (2) Stage-I-Whole, AudioStory regresses bridge queries based on the input caption, *i.e.*, generating $\mathbf{T}_{\text{semantic}}$ and $\mathbf{T}_{\text{residual}}$, respectively. They are subsequently merged via equation 4 and fed into DiT. Here, the regression of $\mathbf{T}_{\text{semantic}}$ and the prediction of its beginning and end tokens are supervised. We tune LoRA of the LLM, all projectors, the attention layer and the generation model DiT. The learning objectives are shown below:

$$\mathcal{L}_{s_1}^{\text{warm}} = \mathcal{L}_{\text{mse}}, \qquad \mathcal{L}_{s_1}^{\text{whole}} = \mathcal{L}_{\text{mse}} + \lambda_1 \mathcal{L}_{\text{text}}^{\text{token}} + \lambda_2 \mathcal{L}_{\text{flow}}, \tag{6}$$

where $\mathcal{L}_{\text{text}}^{\text{token}}$ is only applied to the start and the end tokens of $\mathbf{T}_{\text{semantic}}$. After this Stage-I, AudioStory possesses a strong capability for single-audio generation.

**Stage-II: Single-audio unified generation and understanding.** Building upon Stage-I, we further introduce audio understanding data to enable unified generation and understanding of single-audio clips. The model takes audio as input for understanding. We freeze the audio encoder while the trainable parameters remain the same as Stage-I-Whole. The learning objectives are in Eq equation 7.

$$\mathcal{L}_{s_2} = \mathcal{L}_{\text{mse}} + \lambda_1 \mathcal{L}_{\text{text}} + \lambda_2 \mathcal{L}_{\text{flow}}. \tag{7}$$

With this unified training, AudioStory's generation abilities can be further enhanced.

**Stage-III: Long-audio unified generation and understanding.** We extend the unified training in Stage-II to long-form audio. We further introduce Interleaved Reasoning Generation (Sec. 4.1) with a high-quality multi-audio dataset to perform supervised fine-tuning. For the generation task, the model sequentially infers the number of audio events based on the input instruction, analyzes the audio content, and performs interleaved generation of captions, semantic tokens, and residual tokens. For the audio continuation task, given the input audio and instruction, the model comprehends the inputs, reasons the key events with story details, and finally generates several short audio segments in a clip-by-clip manner. The audio understanding data incorporates audio Q&A and instruction data. We keep the learnable components the same as Stage-II. The overall learning objectives are:

$$\mathcal{L}_{s_3} = \mathcal{L}_{\text{mse}} + \lambda_1 \mathcal{L}_{\text{text}} + \lambda_2 \mathcal{L}_{\text{flow}} + \lambda_3 \mathcal{L}_{\text{reason}}. \tag{8}$$

## 5 EXPERIMENTS

In this section, we first present the experimental setup (Sec. 5.1). Then, we compare AudioStory with existing TTA and unified models on long-form audio generation (Sec. 5.2). We also study the audio understanding and the audio generation (Sec. 5.3) ability of AudioStory in short audio clips, showing its superior fundamental ability. Finally, in Sec. 5.5, we conduct an in-depth exploration of reasoning forms, bridging query types, joint training strategies, and the synergy between understanding and generation, and provide several key insights.

### 5.1 EXPERIMENTAL SETUP

**Implementation details.** We choose Qwen-2.5-3B-Instruct (Yang et al., 2024) as the LLM and employ DiT initialized from TangoFlux (Hung et al., 2024). We employ Whisper-large-v3 (Radford et al., 2023) as the audio encoder for the audio continuation task. The projector has two layers with GeLU activations. In Stage-I, AudioStory is trained with lr= $2e^{-4}$ for 50 epochs with a per-device batch size of 32. In Stage-II, we use lr=1e-4 for 10 epochs. The ratio of understanding and generation data is 2:1. In Stage-III, we set different learning rates for LLM and DiT. We set $\lambda_1 = 1, \lambda_2 = 0.2, \lambda_3 = 0.4$. The tunable parameters three-stage training are LoRAs in LLMs, projectors, the cross-attention fuser for bridging queries, and DiT. More details are in the Appendix.

**Evaluation metrics.** For single-audio generation, we employ Frechet Distance (FD), Frechet Audio Distance (FAD), KL-Divergence (KL), and CLAP score on AudioCaps testset (Kim et al., 2019). For audio understanding, we consider the tasks of audio question answering (AQA), and audio captioning on AudioCaps and Clotho dataset (Drossos et al., 2020), reporting SPIDEr, CIDEr, and ACC scores. The evaluation metrics for long-audio generation are in Sec. 3.

**Baseline methods.** There are two groups: (1) pure TTA models like AudioLDM2 (Liu et al., 2024) and TangoFlux (Hung et al., 2024) and (2) LLM-based unified models, including CoDi (Tang et al., 2023b) and NExT-GPT (Wu et al., 2024). For long-form audio generation, we construct three classes of baselines: (1) Directly generating audios with maximum available durations using the whole textual condition. (2) Incorporating LLMs to reason and generate captions for each short audio clip, which are then fed into TTA models to generate multiple audio clips separately. These clips are then concatenated to constitute the final long-form audio. (3) Directly using the ground truth captions in the benchmark, serving as the oracle setting and upper bounds.

### 5.2 LONG-FORM NARRATIVE AUDIO GENERATION

**Instruction-following ability.** As shown in Table 1, considering the instruction-following aspect, AudioStory demonstrates a significant advantage in complex scenarios involving multiple events and sounding objectives. It outperforms the LLM-aided TTA models by 17.85% on the CLAP score,

Table 1: Comparative results on long-audio generation. "Instruct" is short for instruction-following and "CLAP" denotes CLAP score, "gt" denotes ground-truth. "Consis." and "Coher." are short for consistency and coherence. Here, **bold** and underline indicate the best and the second-best results.

| Model | Instruction-Following | | | Consistency | | Generation Quality | | Max. Duration ↑ |
|---|---|---|---|---|---|---|---|---|
| | Instruct. ↑ | CLAP ↑ | Reasoning ↑ | Consis. ↑ | Coher. ↑ | FD ↓ | FAD ↓ | |
| AudioLDM2 (Liu et al., 2024) | 2.8 | 0.296 | - | 4.6 | 4.4 | 3.43 | 4.49 | 10s |
| TangoFlux (Hung et al., 2024) | 3.2 | 0.317 | - | 4.1 | 4.2 | 2.48 | 3.49 | 30s |
| Caps (gt)+TangoFlux (Hung et al., 2024) | 4.0 | 0.348 | - | 2.4 | 2.0 | 1.79 | 3.59 | 30s |
| LLM+TangoFlux (Hung et al., 2024) | 3.5 | 0.322 | 3.5 | 2.1 | 1.9 | 2.55 | 3.82 | 30s |
| LLM+CoDi (Tang et al., 2023b) | 3.2 | 0.286 | 3.5 | 1.4 | 1.4 | 3.39 | 4.04 | 10s |
| LLM+NExT-GPT (Wu et al., 2024) | 3.3 | 0.299 | 3.5 | 1.8 | 1.7 | 3.47 | 3.99 | 10s |
| AudioStory | **4.1** | **0.392** | **4.2** | **4.1** | **3.9** | **1.43** | **3.00** | **150s** |
| AudioStory-continuation | 4.0 | 0.387 | 4.0 | 4.0 | 3.8 | 1.52 | 3.17 | 150s |

Table 2: Single audio understanding performance.

| Model | ClothoCaps | | ClothoAQA | | AudioCaps | |
|---|---|---|---|---|---|---|
| | SPIDEr | CIDEr | ACC | B-ACC | SPIDEr | CIDEr |
| UIO-2 XXL (Lu et al., 2023) | 5.7 | 6.5 | - | - | - | 48.9 |
| CoDi (Tang et al., 2023b) | 6.2 | 7.3 | - | - | 48.0 | 78.9 |
| NExT-GPT (Wu et al., 2024) | 13.8 | 20.3 | 26.4 | 39.5 | 53.4 | 80.7 |
| Spider (Lai et al., 2024) | - | - | - | - | 53.7 | 81.9 |
| AudioStory-Base | **24.1** | **37.7** | **42.8** | **60.6** | **54.8** | **83.2** |

Table 3: Single audio generation performance.

| Model | AudioCaps Test Set | | | | | |
|---|---|---|---|---|---|---|
| | $FD_{openl3}$ ↓ | $KL_{passt}$ ↓ | FD ↓ | FAD ↓ | KL ↓ | CLAP ↑ |
| Make-An-Audio (Huang et al., 2023) | 128.49 | 1.16 | 1.65 | 3.16 | 0.63 | 0.256 |
| stable-audio-open (Evans et al., 2024) | 103.68 | 1.12 | 1.63 | 2.98 | 0.61 | 0.298 |
| AudioLDM2 (Liu et al., 2024) | 87.74 | 1.01 | 1.59 | 2.63 | 0.57 | 0.252 |
| TangoFlux (Hung et al., 2024) | 83.58 | 0.95 | 1.57 | 2.34 | 0.52 | **0.385** |
| CoDi (Tang et al., 2023b) | 121.66 | 1.17 | 1.69 | 9.61 | 0.60 | 0.228 |
| NExT-GPT (Wu et al., 2024) | 107.18 | 1.13 | 1.64 | 5.69 | 0.59 | 0.265 |
| AudioStory-Base | **83.39** | **0.91** | **1.52** | **2.29** | **0.51** | 0.383 |

thereby demonstrating the superior instruction-following generation capability of our model. Our method effectively addresses the issue of overlooking sounding entities, which can be attributed to the enhanced understanding and decomposition of the instruction.

**Generation quality.** AudioStory demonstrates strong long-form audio generation performance across both natural sound and music domain, outperforming baselines in FD and FAD scores. This improvement stems from: (1) single-clip training, which extends high-quality short-audio generation to longer sequences, and (2) generating longer audio that better matches reference lengths compared to previous methods.

**Consistency.** *Notably, consistency is meaningful only with strong instruction-following.* For example, AudioLDM2, despite high consistency scores from short (10s) outputs, performs poorly on instruction-following, making it a weak baseline. In contrast, our method achieves substantial advantages in both consistency and coherence, reaching scores of 4.0 and 3.7, respectively, as in Table 1. It is worth noting that in the consistency evaluation, AudioStory achieves comparable performance despite generating significantly longer audio with richer narratives compared to TTA models.

## 5.3 SINGLE-AUDIO GENERATION

**Joint audio generation & understanding.** We also evaluate our model's performance on short audio generation and understanding tasks, and conduct comparisons with TTA and LLM-based models. For the generation task in Table 3, AudioStory outperforms prior competitors on both suites of evaluation tools, even outperforming the state-of-the-art TTA model, *i.e.*, TangoFlux (Hung et al., 2024), indicating the effectiveness of the proposed LLM and DiT bridging mechanism. As for the audio understanding task in Table 2, AudioStory outperforms advanced LLM-based models, which means that our method could competently handle both generation and understanding tasks.

## 5.4 QUALITATIVE ANALYSIS

AudioStory exhibits strong reasoning capabilities, it can accurately divide the input instruction into several events based on narrative logic and temporal order, subsequently generating short audio clips segment by segment, and ultimately composing a coherent long-form audio. Besides, AudioStory could accurately infer the duration of each audio clip. Here, we provide a qualitative case in Fig. 3. More cases are presented in the Appendix D.

Figure 3: Qualitative case of long-form audio generation.

## 5.5 ABLATION STUDIES AND ANALYSIS

**Does interleaved reasoning generation help narrative audio generation?** We investigate effective reasoning forms for long-form audio generation, testing two model variants: (a) one that skips instruction analysis, and (b) one without explicitly generating captions for each audio clips. As shown in Table 4, removing reasoning leading to missing audio events, and significantly reduces instruction-following performance. Without interleaved reasoning,

Table 4: Ablations of reasoning.

| Variant | Cons. ↑ | Inst. ↑ | FAD ↓ | CLAP ↑ |
|---|---|---|---|---|
| w/o reasoning | 3.1 | 3.1 | 4.13 | 0.34 |
| w/o interleaved | 1.6 | 1.2 | 16.03 | 0.14 |
| w/ reasoning | **4.0** | **4.1** | **3.06** | **0.39** |

the model infers event content but lacks contextual guidance for generating bridge queries, greatly diminishing audio quality. We conclude that reasoning is indispensable, and explicit captions for each clip are crucial to generation quality.

**Which type of features are suitable for bridging between the LLM and the DiT?** Our analysis shows that audio features, with lower semantic density and greater difficulty for the LLM to interpret, especially due to Whisper's complex temporal structure, are less effective than textual features. Thus, supervising semantic tokens with text is more efficient. For residual tokens, Table 5 (c)–(g) reveals that explicit or weak supervision with audio features harms performance. In summary, textual features are ideal for supervising semantic tokens, while weak supervision via the DiT

Table 5: Ablations on bridging mechanism.

| ID | BQ | Sup. Feat. | Single | Multi |
|---|---|---|---|---|
| (a) | Semantic | AudioMAE (Huang et al., 2022) | 9.55 | 11.39 |
| (b) | | Whisper (Radford et al., 2023) | 10.26 | 12.31 |
| (c) | Residual | AudioMAE (Huang et al., 2022) | 9.24 | 10.06 |
| (d) | | Whisper (Radford et al., 2023) | 11.06 | 11.21 |
| (e) | Residual +guid. | AudioMAE (Huang et al., 2022) | 3.60 | 4.21 |
| (f) | | Whisper (Radford et al., 2023) | 3.71 | 4.39 |
| (g) | Ours | T5 w/o guid. | **2.29** | **3.12** |

loss best captures complementary audio information for residual tokens.

**What are the key factors in end-to-end joint training?** Prior works train LLM and DiT separately, creating a feature gap. We propose end-to-end joint training (Table 6). Notably, removing residual tokens significantly reduces performance, revealing that LLM and DiT focus on different information types, and directly updating the LLM with the DiT loss harms its performance. Residual tokens help mitigate this issue. We also examine DiT's learnable parameters. Fully freezing DiT degrades performance, while full updates yield the best results. Unfreezing MM-DiT outperforms Single-DiT, as the latter focuses on low-level features more sensitive to noise, impacting generation quality. one can draw the following conclusions: (1) End-to-end joint training of the LLM and DiT is essential. (2) Residual tokens capture complementary low-level information and reduce conflicts. (3) Fully unfreezing DiT is necessary; selective unfreezing Single- or MM-DiT leads to suboptimal results.

**How to progressively build the synergy between generation and understanding?** We evaluate the effectiveness of various training aspects. Table 7 shows that without progressive training, both comprehension and generation significantly decline due to their inherent conflict. In contrast, a structured progressive strategy enables unified training to outperform isolated approaches. Training generation first, followed by comprehension, achieves the best overall performance with strong comprehension accuracy. Reversing the order harms generation, while interleaved training also undermines optimization. We conclude that generation and comprehension have inherent synergy, with the optimal training order depending on the primary objective.

Table 6: Ablations on the end-to-end joint training strategy of DiT. Here "S-DiT" and "M-DiT" denote Single-DiT and MM-DiT. "Consis." denotes consistency.

| ID | Semantic Tokens | Residual Tokens | DiT Joint Training | Tunable Module | Single Audio | | | Multi Audio | |
|---|---|---|---|---|---|---|---|---|---|
| | | | | | FD ↓ | FAD ↓ | KL ↓ | Consis. ↑ | FAD ↓ |
| (a) | ✓ | ✗ | ✗ | - | 1.57 | 2.33 | 0.52 | 3.2 | 5.23 |
| (b) | ✓ | ✗ | ✓ | open all | 2.16 | 4.66 | 0.84 | 3.4 | 4.98 |
| (c) | ✓ | ✗ | ✓ | freeze | 4.86 | 11.04 | 0.89 | 1.3 | 12.97 |
| (d) | ✓ | ✓ | ✓ | open S-DiT | 2.37 | 5.84 | 0.64 | 2.1 | 6.28 |
| (e) | ✓ | ✓ | ✓ | open M-DiT | 1.98 | 3.21 | 0.67 | 3.5 | 3.64 |
| (f) | ✓ | ✓ | ✓ | open all | **1.53** | **2.29** | **0.51** | **4.3** | **3.00** |

Table 7: Ablations on progressive training. "Gen.", "Und." and "BQ" denote generation, understanding and Bridge Queries. "SAG" and "LAG" are short for single and long-form audio generation.

| ID | Order | Stage-I | Stage-II | Stage-III | SAG | LAG | Audio Und. | |
|---|---|---|---|---|---|---|---|---|
| | | | | | FAD ↓ | FAD ↓ | CIDEr ↑ | SPIDEr ↑ |
| (a) | | Und. | - | - | - | - | 35.7 | 23.1 |
| (b) | Und.→Gen. | Und. | BQ | - | 7.42 | 9.53 | 36.9 | 23.8 |
| (c) | | Und. | BQ | DiT joint | 6.50 | 7.26 | **38.6** | **24.9** |
| (d) | | BQ | - | - | 2.37 | 5.23 | - | - |
| (e) | Gen.→Und. | BQ | Und. | - | 2.35 | 4.98 | 31.5 | 19.5 |
| (f) | | BQ | Und. | DiT joint | 3.61 | 6.50 | 24.6 | 16.4 |
| (g) | | BQ | DiT joint | Und. | **2.29** | **3.00** | 37.7 | 24.1 |
| (h) | N/A | DiT joint + Und. | | | 5.70 | 8.74 | 27.3 | 18.2 |

Table 8: Human evaluation of the generated audios for methods on instruct-following, consistency, fidelity, and reasoning logic.

| Method | Instruct-Follow | Consist. | Fidelity | Reason. Logic |
|---|---|---|---|---|
| LLM + TangoFlux | 3.52 | 3.22 | 3.58 | 3.19 |
| LLM + NExT-GPT | 3.10 | 2.56 | 2.87 | 3.14 |
| AudioStory (Ours) | 4.23 | 4.68 | 4.37 | 4.22 |

Table 9: Correlation of Gemini and human scores.

| | Across model | Across model |
|---|---|---|
| Kappa Coef. | 0.91 | 0.83 |

## 5.6 HUMAN EVALUATION

Beyond API-based evaluation, we conducted an anonymous user study with 30 participants manually scoring 150 long-form narrative audio clips from 50 instructions across three methods. As shown in Table 8, AudioStory consistently outperforms competitors in instruction-following, consistency, quality, and reasoning. We compute Cohen's kappa to measure agreement across methods and samples, with results in Table 9 showing strong alignment between human and Gemini scores, confirming the reliability of the Gemini-based evaluation. Further details are provided in Appendix C.

## 6 CONCLUSION

In this paper, we tackle the key limitations of existing methods in generating long-form narrative audio in complex scenarios. We introduce AudioStory, a unified understanding-generation model endowed with robust multimodal instruction-following and reasoning. To achieve this, we design an interleaved reasoning generation process, a decoupled bridging mechanism, and a progressive training strategy. Additionally, we present AS-10k, the first benchmark for long-form narrative audio generation, which includes fine-grained annotations of audio and audio-visual events and detailed reasoning trajectories. Our comprehensive analyses cover reasoning forms, bridge query types, end-to-end training strategies for LLM-DiT integration, and the collaborative dynamics between understanding and generation, providing practical insights for future model development.

**Limitations and Future Work.** Our work primarily targets the natural sound and music domains, which require further research. Future efforts will explore incorporating speech, aiming for a unified model across all auditory domains. Moreover, since multimodal instruction for long audio generation is still underexplored, future work can integrate more sophisticated designs, such as using multiple audio generators to address overlapping audio segments. We also plan to blend text and audio generation within the same autoregressive multimodal LLM.

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

# APPENDIX

## A  IMPLEMENTATION DETAILS

We provide detailed hyper-parameters of three training stages in Table 10. In Stage-II and Stage-III, the ratio of generation and understanding samples is 2:1. For LLM, we choose Qwen2.5-3B-Instruct and only tune LoRA to avoid overfitting. TangoFlux is employed as the initialization of DiT for audio generation. For the weights of different loss functions, we set the weight of $\mathcal{L}_{\text{mse}}$ for T5 regression, $\mathcal{L}_{\text{text}}$ for next-token-prediction and $\mathcal{L}_{\text{flow}}$ for DiT as 5, 2 and 1, respectively.

Table 10: Detailed hyper-parameters of three training stages. Here, "A" denotes audio, "proj." and "lr" are short for the projector and learning rate. We use 16 GPUs and report the overall batch size.

| Dimension | | Stage-I | | Stage-II | Stage-III |
|---|---|---|---|---|---|
| | | Warm-up | Whole | | |
| Task | | A→T5 | A→T5 with DiT. | A→T5 with DiT + Und. | A→T5 with DiT + Und. + Reasoning |
| Dataset | | AudioCaps, WavCaps | | I+AudioSetCaps (Q&A), VGGSound (Q&A), MusicCaps, Auto-ACD | AS-10k |
| Model | Trainable | LLM, proj. ($\mathbf{T}_{\text{semantic}}$) | LLM, all proj., DiT | LLM, all projectors, DiT | LLM, all proj., DiT |
| | Frozen | Whipser, DiT | Whisper | Whisper | Whisper |
| Training Config | batch size | 512 | 256 | Gen.: 8, Und.: 16 | Gen.: 8, Und.: 16 |
| | lr | 1e-3 | | 1e-4 | LLM (2e-5), DiT (5e-5) |
| | epoch | 25 | 25 | 10 | 10 |

## B  TRAINING DATASETS

The training dataset comprises the understanding dataset, single-audio generation and multi-audio (long-audio) generation datasets. For the understanding dataset, we integrated AudioSetCaps (Gemmeke et al., 2017), VGGSound (Chen et al., 2020), MusicCaps (Agostinelli et al., 2023), and Auto-ACD (Sun et al., 2024), converting their captions into QA format. Additionally, we incorporated AudioSetCaps-QA and VGGSound-QA datasets, resulting in 1M audio-QA pairs in total. For the single-audio generation dataset, we combined AudioSetCaps, VGGSound, MusicCaps (Agostinelli et al., 2023), and Auto-ACD, resulting in 700k audio-caption pairs. For the multi-audio generation dataset, we curated the AS-10k dataset, with details provided in Sec. 3. In Stage-I, we train the model on we train the model on single-audio generation datasets. Stage-II further incorporates the audio understanding dataset beyond Stage-I. As for Stage-III, our model is trained using multi-audio generation as well as understanding datasets.

## C  HUMAN EVALUATION

**Evaluation protocol.** Beyond API-based evaluation, we further conducted an anonymous user study on our model and baseline models. We employ 30 participants to manually score a total of 150 audio clips, generated from 50 instructions, by our model, Tangoflux, and Next-GPT, respectively. The participants listened to the long-form audio generated by different models based on the same instruction. They scored the audio on four criteria: instruction-following, consistency, generation quality, and reasoning logic. The scores were averaged to compute user consistency. As shown in the Table 8, AudioStory consistently outperforms other competitors in terms of instruction-following, consistency, quality and reasoning logic.

**Correlation between Gemini-based & human-based evaluation.**  Qualitatively, human evaluation results show our model performs the best among all three models, with the LLM + TTA model outperforming the LLM + any-to-any model. This aligns with the results from our Gemini evaluation. Quantitatively, we analyze the correlation between the human subjective and Gemini-based objective evaluation. We calculate Cohen's kappa coefficient between these two evaluation protocols. Specifically, we compute the correlation across two dimensions, *i.e.*, different comparative methods

Figure 4: Case of naive video dubbing: First, we extract captions from the video, then write the extracted captions as instructions and send them to AudioStory for audio generation.

in Table 8 and different test samples. The results in Table 9 indicate a high correlation between the human and Gemini scoring distributions across various models and samples, validating the correctness of the proposed Gemini-based evaluation.

## D  MORE QUALITATIVE CASES

**Instructional long-form audio generation and continuation.** We present more cases for long-form audio generation. Our model could automatically derive the duration of each audio segment to be generated, as shown in Fig. 5, Fig. 6 and Fig. 7. One could observe that AudioStory could accurately determine the number of events based on the instruction and provide precise descriptions for each audio segment, including both the events themselves and their associated sound effects. Finally, AudioStory can precisely reason out the caption for each segment and generate the corresponding audio clips accordingly.

**The case of Tom&Jerry.** Given an instruction for generating music of Tom&Jerry style, our model can generate sound that not only aligns with the musical style of the Tom&Jerry cartoon but also vividly reflects the characters' emotions and storyline, exhibiting strong generalization capabilities.

## E  EXTENDED APPLICATIONS

**Video dubbing.** While previous experiments have focused on text-based instructions, we now extend the application of AudioStory to a more practical scenario: video dubbing. This enhancement enables the model to thoroughly analyze video content, reason about the sequence of events and

---

### Instructional Audio Generation

📝 **Instruction：**

Construct a detailed soundscape that tells the story of a young beatboxer performs energetically outdoors, introduces themselves as 'yoh!', and requests tips. The total duration is 41.8 seconds.

input 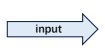 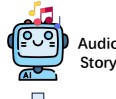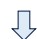 Audio Story

💡 **Generated Reasoning：**

The caption can be divided into four points:\n\n1. [0.0, 2.5]\n **Story Details:** The beatboxing begins indoors and transitions outdoors with a performer giving a shout.\n **Sound Effects:** Energetic beatboxing, voice saying \"Yoh!\", camera movement indicating a change in location.\n\n2. [2.5, 21.5]\n **Story Details:** A young man starts beatboxing complex rhythmic patterns, intensifies with \"skrrt\", and ends with a punchline.\n **Sound Effects:** Complex rhythmic beatboxing, intense \"skrrt\", loud breaths, informal ending statement.\n\n3. [21.5, 27.5]\n **Story Details:** Audience responds with laughter and brief verbal appreciation, and the performer clarifies in a friendly manner.\n **Sound Effects:** Laughter, verbal appreciation, friendly clarification in response to an audience member's request.\n\n4. [27.5, 41.8]\n **Story Details:** Performer thanks the audience, introduces as Yoh\", asks for tips, and expresses excitement.\n **Sound Effects:** Grateful thank-you, friendly self-introduction, direct question, background noise from phone call.

🎵 **Generated Caption：**

- Energetic beatboxing performance outdoors, abruptly shifting to a voice shout of \"Yoh!\" as the camera pans, indicating a move outside. <timestamp>2.5</timestamp>
- A young man enthusiastically beatboxes, creating complex and rapid rhythmic patterns that intensify with sharp 'skrrt' sounds, followed by loud, audible breaths, ending with a punchline-like statement. <timestamp>19.0</timestamp>
- Audience laughter followed by a brief, appreciative word from the performer, \"Okay.\" <timestamp>6.0</timestamp>
- The performer thanks the audience, introduces themselves as 'Yoh!' and explains they don't have a YouTube channel, then asks for tips directly in a friendly tone. <timestamp>14.3</timestamp>

Figure 5: Long-form audio generation case #2.

---

### Instructional Audio Generation

📝 **Instruction：**

Construct a detailed soundscape that tells the story of an instructor demonstrates common push technique errors in table tennis, emphasizing power, wrist angle, and swing size issues. The total duration is 43.6 seconds.

input 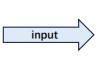 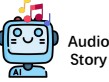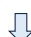 Audio Story

💡 **Generated Reasoning：**

The caption can be divided into 4 points.\n\n1. [0.0, 11.8]\n**Story Details**: Instructor demonstrates a weak push technique leading to hitting the net.\n**Sound Effects**: Instructor's voice explaining the issue, followed by a ball hitting the net.\n\n2. [11.8, 24.8]\n**Story Details**: Instructor emphasizes the need for an upright wrist during the push.\n**Sound Effects**: Consistent, fast ball hitting sounds while demonstrating wrist movement.\n\n3. [24.8, 37.5]\n**Story Details**: Demonstrates a weak push by hitting the table and foot repeatedly.\n**Sound Effects**: Loud ball hitting sounds, followed by a ball hitting the foot, and repeated crashing on the table.\n\n4. [37.5, 43.6]\n**Story Details**: Fails misery by hitting the ball with the paddle.\n**Sound Effects**: Instructor exclaiming an error, single dramatic, high-pitched paddle-ball impact sound.

🎵 **Generated Caption：**

- An instructor explains that pushing this way lacks power and emphasizes an upright wrist technique to prevent the ball hitting the net. The ball does not.go in the hole effectively. <timestamp>11.8</timestamp>
- The instructor discusses the importance of using an upright wrist during the execution of a push, emphasizing how it affects the ball's direction while demonstrating hand movement slowly at a consistent speed. <timestamp>13.0</timestamp>
- The instructor repeats a sequence three times, ending with a weak push. A loud banging sound from the ball on the table, followed by a dull thud where the ball hits the foot, accompanied by repeated crashing sounds. <timestamp>12.7</timestamp>
- The instructor exclaims with surprise and amusement, \"Oh my god, you hit the ball with the paddle!\" followed by a single, dramatic, high-pitched paddle-ball impact sound. <timestamp>6.1</timestamp>

Figure 6: Long-form audio generation case #3.

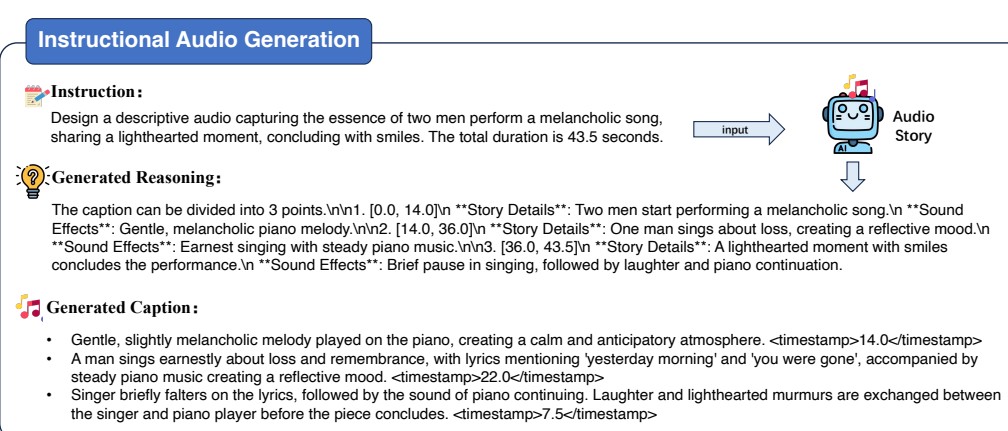

Figure 7: Long-form audio generation case #4.

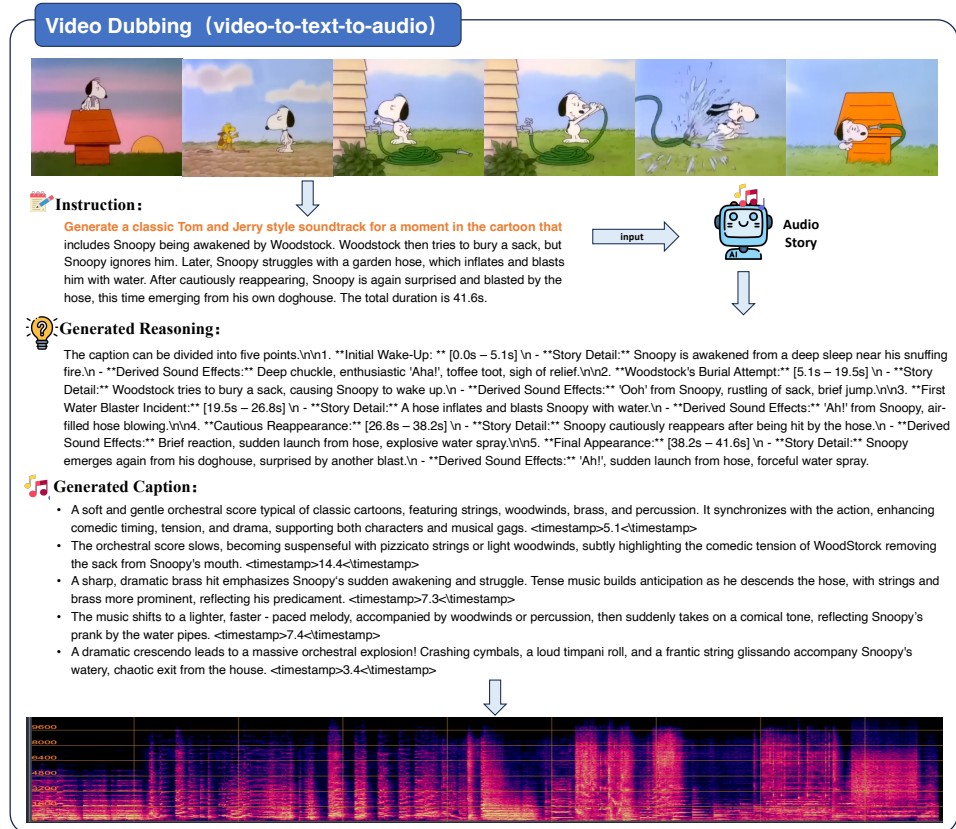

Figure 8: Case of naive video dubbing: First, we extract captions from the video, then write the extracted captions as instructions and send them to AudioStory for audio generation.

their corresponding timestamps, and generate synchronized audio. An initial approach is to employ Gemini-2.5-pro to generate a caption summarizing the entire video, followed by instruction-based audio generation, as illustrated in Fig. 8. Specifically, given the video without audio, we first generate the visual captions and convert them into the form of instructional language. These instructions are subsequently fed into our model, *i.e.*, AudioStory, to generate the audio. As a whole, we achieve video dubbing in this multi-step process, *i.e.*, video→visual caption→instruction→audio. Here, we provide a case of `Snoopy`. We use our model AudioStory trained for Tom&Jerry. As in Fig. 9,

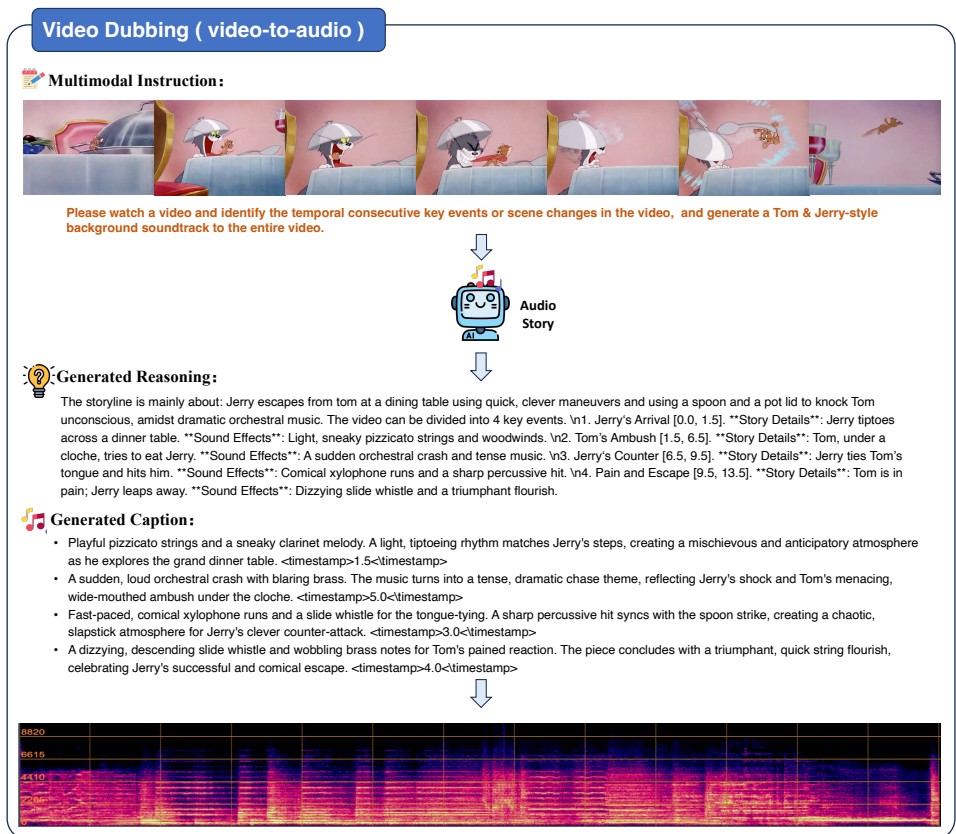

Figure 9: Case of video dubbing: We input both the video and the instruction into the model, which parses the narrative into segments, extracts story details with corresponding audio elements, and sequentially generates aligned audio clips.

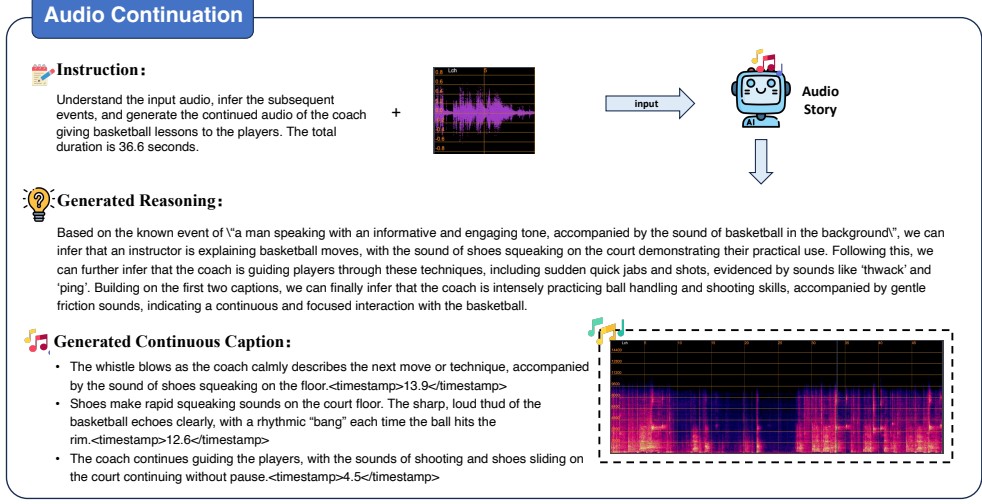

Figure 10: Qualitative cases of audio continuation #1.

the video is divided into four distinct segments, with the generated audio closely aligning with the Tom&Jerry style, effectively reflecting Snoopy's emotions, *e.g.*, the calmness of waking up, the surprise while playing with the water pipes, and the humorous tone at the end. Notably, for any given

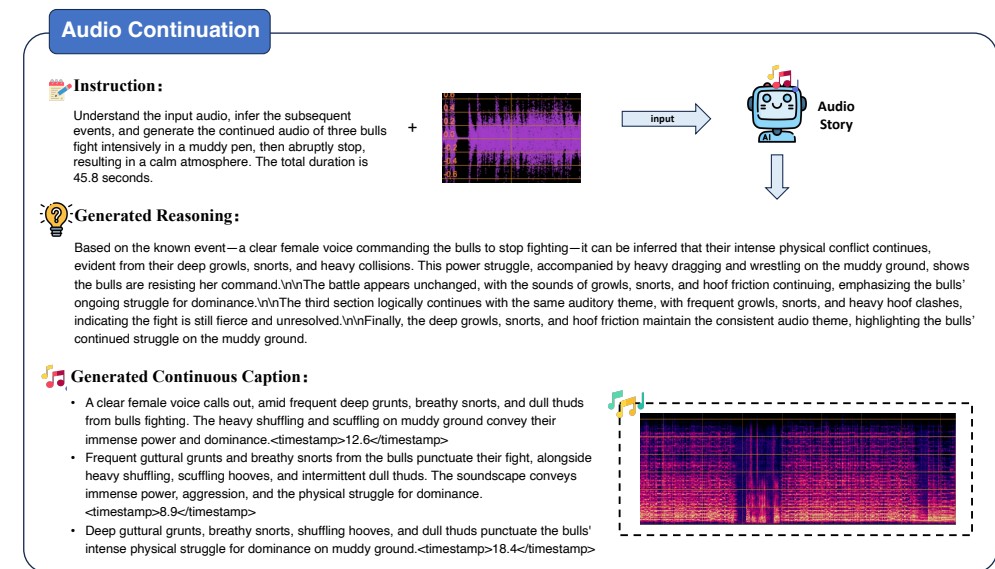

Figure 11: Qualitative cases of audio continuation #2.

video, AudioStory could generate soundtracks that match the musical characteristics of Tom&Jerry in a zero-shot manner, which is a unique and interesting application of our model.

However, this method is not conducive to producing audio that aligns closely with the visual content. Furthermore, the model is designed to accept both video data and instructions as input. The LLM performs reasoning on the video and produces bridging tokens. During the reasoning phase, the LLM first understands the overall content of the video, then sequentially breaks it down into events based on their temporal order. It infers the specific visual details and corresponding audio information for each event. Technically, we replace the LLM with a pretrained video MLLM (*i.e.*, Qwen2.5-VL (Bai et al., 2025)) and jointly train the LLM and audio generator using LoRA tuning. The training data is from the animated sound partition of AS-10k. We provide the video dubbing results in Fig. 9.

**Audio continuation.** Given an audio segment and an instruction, our model performs audio continuation. AudioStory first reasons about the content of the subsequent audio to be generated, then proceeds with segment-by-segment generation. The concatenated results are shown in Fig. 10 and Fig. 11.

## F  MORE EXPLORATIONS OF RESIDUAL TOKENS

For residual tokens, we not only explore their forms and training strategies, but also investigate hyperparameters such as their quantity and fusion methods with semantic tokens.

**The number of residual tokens.** Here, we study the impact of different numbers of residual tokens, and report both single- and long-form audio generation, as in Table 11. For single-audio generation, too few residual tokens lead to degraded performance. We attribute this to two factors: less low-level complementary information is captured. Additionally, residual tokens help mitigate conflicts between the LLM and the DiT, while too few tokens weaken this effect. Conversely, an excessive number of tokens also degrades performance, because they increase the difficulty for the LLM to regress. Similar patterns could also be observed in the long-form scenario. Overall, 8 residual tokens are most suitable for both single and long audio scenarios.

**Merging mechanism of residual tokens.** For the merging mechanism between semantic and residual tokens, we also conduct in-depth explorations. Here, we mainly consider concatenation and cross-attention. The results of long-form audio generation are reported in Fig. 12. From the results, one can observe that compared to concatenation, cross-attention ensures more effective fusion of the

Table 11: Detailed ablations of the number of residual tokens

| # Tokens | Single Audio | | | Long Audio |
|---|---|---|---|---|
| | FD ↓ | FAD ↓ | KL ↓ | Consistency ↑ |
| 1 | 4.01 | 5.02 | 0.93 | 3.2 |
| 4 | 3.64 | 3.95 | 0.96 | 3.9 |
| 8 | **1.53** | **2.29** | **0.51** | **4.1** |
| 16 | 3.51 | 3.75 | 0.94 | 3.9 |

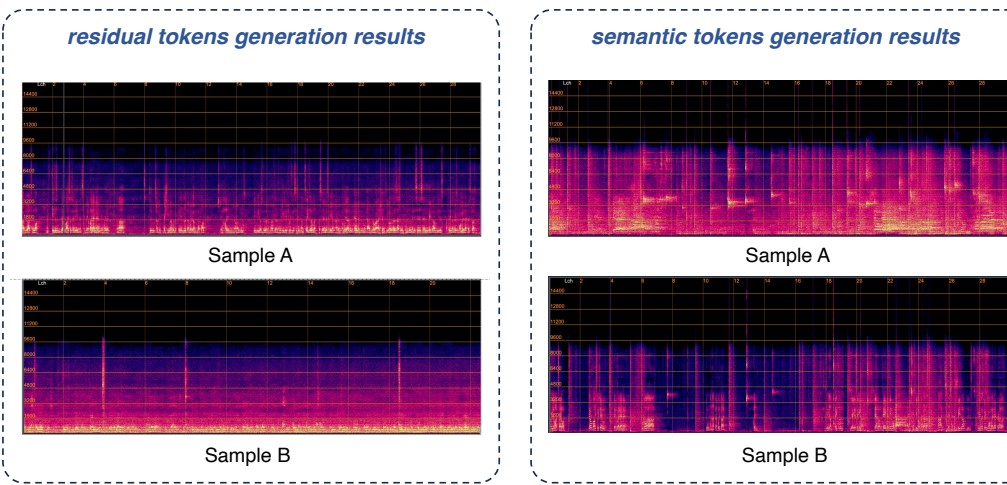

Figure 13: Visualizations of residual tokens.

two features. Additionally, zero-initializing the final layer of the cross-attention module is necessary to prevent excessive disturbance to the semantic tokens at the beginning of training.

## G  WHAT DO RESIDUAL TOKENS LEARN?

To thoroughly explore the effect of residual tokens, we provide visualizations in Fig. 13 (left). Specifically, the DiT takes *only* the residual tokens as the input and generates its corresponding audio. We subsequently concatenate all audio clips to constitute the whole long-form audio. The results reveal that for the same audio sample, the residual tokens capture temporally consistent low-level information, primarily reflecting coherence across different audio clips. In contrast, for different samples, the learned residual characteristics vary distinctly. By contrast, semantic tokens learn the underlying global semantics of the input audio and represent the progression of events over time, as illustrated in Fig. 13 (right).

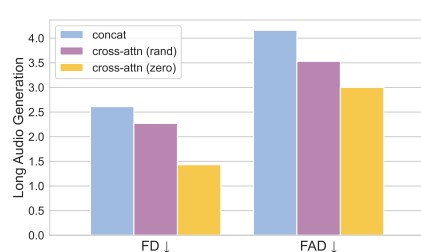

Figure 12: Ablations of token merging.

## H  AS-10K BENCHMARK

### H.1  DATASET CONSTRUCTION PIPELINE

The dataset construction pipeline is illustrated as follows. First, we filter videos to select those containing continuous audio events with visually grounded storylines. Next, in the event parsing

---

**Annotation prompt**

**UnAV annotation prompt**

Please watch a video clip and provide a detailed analysis of the video, aiming to generate high-quality input for subsequent text-to-speech synthesis. The results should be returned in JSON format and should include the following sections: (1) Main Event Analysis: Identify key events or scene changes in the video. Output the event's timestamp (start and end timestamps, two decimal places) and description. The description should integrate visual and audio information, vividly depicting the scenario and feelings at the time the event occurs. (2) Sound Effect Analysis: Identify sound effects (non-musical) in the video. Output the timestamp (start and end timestamps, two decimal places), type (e.g., footsteps, door closing, bird chirping), sound_description (description of the sound), and visual_context (description of the related visual scene) for each sound effect. The sound_caption must be detailed, vivid, and expressive, capturing the characteristics, quality, source, intensity, duration of the sound, and any emotions or environmental information it may convey, to facilitate high-quality sound synthesis. The visual_context should succinctly describe the visual imagery directly related to the sound effect. (3) Music Analysis: Identify background music or significant musical segments in the video. Output the timestamp (start and end timestamps, two decimal places) , type (style or genre), sound_description (description of the sound), and visual_context (description of the related visual scene) for each music segment. The sound_caption should detail the characteristics of the music, including melody, rhythm, instruments used, emotional atmosphere, and how it complements the visuals, also aimed at high-quality sound synthesis. The visual_context should succinctly describe the visuals present when the music occurs.

**Tom & Jerry annotation prompt**

Please watch a video clip of 'Tom and Jerry' and provide a detailed analysis of the video, aiming to generate high-quality input for subsequent text-to-speech synthesis. The results should be returned in JSON format and should include the following sections: (1) Main Event Analysis: Identify key events or scene changes in the video. Output the event's timestamp (start and end timestamps, two decimal places) and description. The description should integrate visual and audio information, vividly depicting the scenario and feelings at the time the event occurs. (2) Sound Effect Analysis: Identify sound effects (non-musical) in the video. Output the timestamp (start and end timestamps, two decimal places), type (e.g., footsteps, door closing, bird chirping), sound_description (description of the sound), and visual_context (description of the related visual scene) for each sound effect. The sound_caption must be detailed, vivid, and expressive, capturing the characteristics, quality, source, intensity, duration of the sound, and any emotions or environmental information it may convey, to facilitate high-quality sound synthesis. The visual_context should succinctly describe the visual imagery directly related to the sound effect. (3) Music Analysis: Identify background music or significant musical segments in the video. Output the timestamp (start and end timestamps, two decimal places) , type (style or genre), sound_description (description of the sound), and visual_context (description of the related visual scene) for each music segment. The sound_caption should detail the characteristics of the music, including melody, rhythm, instruments used, emotional atmosphere, and how it complements the visuals, also aimed at high-quality sound synthesis. The visual_context should succinctly describe the visuals present when the music occurs.

Figure 14: AS-10k annotation prompts of Gemini-2.5-pro.

stage, we use Gemini-2.0-flash to decompose each video into multiple key audio events, each annotated with a timestamp, audio caption, and visual caption, as in Fig. 14. Finally, we perform instruction generation: based on fine-grained textual annotations, GPT-4o is used to generate diverse narrative instructions, accompanied by reasoning steps including task decomposition, audio event timeline planning, scene transitions, and emotional tone inference.

## H.2 BENCHMARK CONSTRUCTION

**Dataset prompt.** The constructed dataset consists of instructions, reasoning, and audio clips, each with its caption and duration. Specifically, after parsing videos into key audio events using Gemini-2.0-flash as described in Sec. 3, we obtain annotations for each event including timestamps, audio captions, visual captions, and audiovisual event captions. For instruction generation, we use audio-visual event captions as the source input. A prompt, shown in Fig. 15, is used to summarize the whole caption of the full audio, which is then incorporated into a predefined instruction template to produce the final instruction. For reasoning generation, we provide GPT-4o with the whole caption along with the individual captions for each audio clip. GPT-4o is then prompted to infer the reasoning structure. The reasoning consists of two levels: a high-level decomposition indicating how the whole caption can be divided into several parts, followed by detailed descriptions for each part, including the corresponding events and sound-producing content. An example is illustrated in Fig. 16.

**Annotation prompt**

**instruction prompt**

I will provide you with "Main Event Analysis", which has multiple descriptions in chronological order. Please combine the captions in each description while considering the timestamp information, and summarize them into a brief story summary, namely whole caption. Note: Only output the final whole caption within 20 words.

**reasoning prompt**

I will provide you with an input in JSON format, containing a "Sound Analysis" with multiple "sound description" in chronological order. Meanwhile, I will also provide you with a "whole_caption", representing the complete storyline summarized from these descriptions.

Imagine that you are given only the whole caption and required to generate a detailed story description. You need to output the reasoning process. When doing so, you should not directly rely on the given "sound_description". You can make reasonable inferences and deduce the details of the story events and details, then infer the sounds objects and events, ensuring that the expanded sound details align with the given "sound description".

The reasoning process could be divided into several points (same as the number of given "sound description" in "Sound Analysis"). The format MUST be: first, state how many points the caption can be divided into, and then elaborate on each point. Each point should follow the reasoning style explicitly with clear inferences and logical conjunctions, and state the story details and the derived sound effects. Each point MUST be within 15 words.

Figure 15: AS-10k annotation prompts of GPT-4o.

**AS-10k Cases**

**Instruction**

I'd like an extended audio track with rich details of a soulful singer receives loud applause and praise from the audience and a man, celebrating her powerful performance. The total duration is 35.5 seconds.

**Reasoning**

The caption can be divided into 3 points:\n\n1. **[0.0, 11.5]**\n - **Story Details**: The singer delivers a powerful and soulful performance.\n - **Sound Effects**: Intense singing, live band music.\n\n2. **[11.5, 21.5]**\n - **Story Details**: Performance ends; audience erupts into applause and cheers.\n - **Sound Effects**: Applause, cheers.\n\n3. **[21.5, 35.5]**\n - **Story Details**: A man joins, congratulates, praises the singer on stage; audience continues cheering.\n - **Sound Effects**: Amplified voice of man, applause, cheers.

**Audio Clips**

**Caption 1**

A young woman delivers a powerful and soulful vocal performance, singing with intense emotion and impressive vocal runs, accompanied by a live band including a keyboardist and a guitarist.<timestamp>11.5 </timestamp>

**Caption 2**

The music stops abruptly as the singer hits her final note. The small audience erupts into loud, enthusiastic applause and cheers, filling the venue with sounds of appreciation .<timestamp>10.0 </timestamp>

**Caption 3**

The man joins the singer on stage, congratulates her with an amplified voice, praises her talent. The audience claps and cheers. The atmosphere is celebratory and appreciative .<timestamp>14.0 </timestamp>

Figure 16: AS-10k dataset cases.

**Benchmark evaluation.** Along with the curated dataset, we also construct the long-form narrative audio generation task and its associated benchmark.

(1) Evaluation with Gemini-2.0-flash API, assessing consistency, coherence, instruction following, and reasoning logic. (2) Evaluation with traditional metrics to measure audio generation quality, including FD, FAD, and CLAP score, among others.

For the Gemini-based evaluation, we design tailored scoring criteria for each metric:

**(a) Consistency.**

- **Timbre and Sonic Cohesion** Evaluate whether the primary sound sources maintain a generally consistent timbre and unified sonic characteristics.

- **Sound-Producing Entity Consistency** Assess whether the implied sound-producing entities remain consistent, or if changes feel natural and logical within the audio.

- **Acoustic Environment Consistency** Evaluate the background ambience, reverberation, and spatial impression for overall consistency or reasonable progression.

- **Transition Smoothness** Assess whether the transitions between segments are smooth and free of jarring disruptions..

**(b) Coherence.**

- **Intentional Transitions** Check whether transitions between segments are smooth, purposeful, and naturally connected.

- **Dynamic and Emotional Flow** Assess if the dynamic and emotional progression feels consistent or evolves logically, without unjustified sudden shifts.

- **Tempo and Textural Compatibility** Evaluate whether tempo, rhythm, and sonic textures between segments are compatible and blend cohesively.

- **Transition Smoothness** Judge if segment connections are fluid, without abrupt or disjointed

**(c) Instruction following.**

- **Overall Semantic Alignment** Evaluate whether the generated audio broadly reflects the intended scene, actions, and atmosphere described in the instruction. Minor differences are acceptable if the main idea remains clear.

- **Key Element Presence** Verify whether the important sound-producing entities, actions, and environmental elements mentioned in the instruction are reasonably represented. Missing a few non-central elements is acceptable if key parts are present. Additional sounds not specified in the instruction are acceptable if they logically fit the scene and do not disrupt coherence.

- **Event Sequence and Logical Development** Assess whether the overall event progression is reasonable according to the instruction. Small deviations in order are acceptable if they do not break the logical flow.

- **Specific Sound Detail Accuracy** Evaluate whether important sound features (such as types of sounds, tonal qualities, or intensities) are reasonably reflected. Natural variations are acceptable as long as they do not change the overall character of the audio.

**(d) Reasoning logic.**

- **Overall Reasoning Logic** Evaluate whether the model demonstrates a coherent, logical process in interpreting the instruction and planning the audio scene.

- **Caption-Instruction Alignment** Assess whether the generated audio caption accurately reflects the instruction's key content, sound-producing elements, and described environment.

- **Event Coverage Completeness** Determine whether the inferred and described audio events fully cover the instruction's core elements, with no major omissions.

- **Semantic and Temporal Accuracy** Evaluate whether the implied timeline and semantic structure of the generated audio align with the instruction's flow and intent.

## H.3 SINGLE-AUDIO EVALUATION DETAILS

To evaluate the audio generation model, four key metrics assess different aspects of performance:

- Frechet Distance (FD) measures the statistical similarity between log-Mel spectrogram distributions of generated and real audio, quantifying low-level spectral fidelity (*e.g.*, pitch, timbre) through mean and covariance comparisons in the mel-spectral domain.

- Frechet Audio Distance (FAD) extends FD using high-level embeddings from a pre-trained audio encoder (*e.g.*, VGGish), evaluating perceptual and semantic realism by comparing abstract features like instrument timbre, musical structure, and environmental acoustics.

- CLAP Score calculates the cosine similarity between audio and text embeddings from a cross-modal model, assessing how well generated audio aligns with semantic prompts (*e.g.*, textual descriptions of sound content or context).

- KL-Divergence (KL) measures the distributional dissimilarity between generated and real audio features (spectral, latent, *etc*.), identifying consistency in probability distributions and helping debug issues like mode collapse or over-dispersion in outputs. Collectively, these metrics ensure a comprehensive evaluation of spectral realism, perceptual quality, semantic accuracy, and distributional consistency in generated audio.

