# OpenReview forum: "AudioStory: Generating Long-Form Narrative Audio with Large Language Models"
_ICLR.cc/2026/Conference — ICLR 2026 Conference Withdrawn Submission_

### Official Review · Reviewer_2sNv · 2025-10-16

**Soundness:** 3
**Presentation:** 3
**Contribution:** 3
**Rating:** 6
**Confidence:** 4

**Summary:**

Paper introduces framework for generating long-form narrative audio by integrating large language models (LLMs) with text-to-audio (TTA) systems, addressing challenges such as temporal coherence and compositional reasoning, It employs LLMs to decompose complex instructions into structured sub-tasks, ensuring smooth scene transitions and emotional tone consistency. ​ Specifically, a decoupled bridging mechanism is proposed for semantic and residual tokens for improved audio fidelity and temporal alignment, and an end-to-end progressive training strategy that synergizes instruction comprehension and audio generation. ​ The authors also present AS-10k, the first benchmark for narrative audio generation, featuring diverse annotated audio clips from natural and animated soundscapes. ​ Experiments demonstrate AudioStory's compliance in instruction-following, audio fidelity, and consistency compared to existing models. ​ Applications include instructional audio generation, video dubbing, and audio continuation. ​

**Strengths:**

1. The use of semantic and residual tokens improves audio fidelity and temporal consistency, enabling seamless collaboration between LLMs and diffusion models. ​

2.  The progressive training is end-to-end which enhances synergy between instruction comprehension and audio generation, eliminating the need for modular pipelines. ​

3. The paper establishes AS-10k, the first benchmark for narrative audio generation, with diverse annotated audio clips from natural and animated soundscapes. ​

4. Extensive experiments show AudioStory's high performance in instruction-following, audio fidelity, and consistency compared to existing models. ​

**Weaknesses:**

1. Two T_residual appear in Equation (4), is one of them redundant?

2. For the evaluation, what’s the duration of the generated audio being evaluated?

3. While the author claims to have video dubbing ability, there’s no quantitative evaluation on this. Also, since there’s no fine-grained temporal condition within each event segment, there will be missing-out audio for some actions.

4. I think the largest limitation is that the framework primarily targets natural sound and music domains, leaving out other auditory domains like speech. ​

5. The proposed framework involves intricate mechanisms, such as decoupled bridging tokens and progressive training strategies, which may pose challenges for replication and practical implementation. ​

6. Limited overlapping audio capabilities.

7. Video dubbing application is described as a multi-step process that may not align closely with visual content.

8. Human dubbings are inaudible.

9. Evaluation relies heavily on Gemini-based scoring, which may introduce bias or limit generalizability without further validation. ​

**Questions:**

Note to authors: the preliminary rating would’ve been more like 5 (not allowed) which will be finalized during/after rebuttal.

1. How well does the current AudioStory generalize to other auditory domains, such as speech or conversational audio, beyond natural sounds and music?

2. How well the current AudioStory handle overlapping audio: any failure cases?

3. How does the model perform in terms of computational efficiency and scalability when generating very long audio sequences (e.g., several minutes or hours)?

4. How can the video dubbing process be improved to ensure tighter alignment between visual content and generated audio?

5. How reliable is the Gemini-based evaluation compared to human evaluation, and are there plans to incorporate additional objective metrics for validation? ​

6. Are there any biases or limitations in the AS-10k dataset that could affect the model's performance or generalizability?

**Details Of Ethics Concerns:**

No ethics concerns.

---

### Official Review · Reviewer_2CMy · 2025-10-21

**Soundness:** 3
**Presentation:** 2
**Contribution:** 2
**Rating:** 2
**Confidence:** 4

**Summary:**

This paper presents AudioStory, a reasoning-guided text-to-audio framework that decomposes narrative prompts into temporally ordered events using an LLM and generates corresponding audio segments via a DiT-based diffusion model. The method supports multimodal instructions (text, audio, or video) and introduces the AS-10k dataset for training and evaluation.

**Strengths:**

1. The decoupled bridging mechanism provides an interesting formulation for linking LLM reasoning with audio diffusion models.

2. The introduction of AS-10k represents a valuable benchmark contribution to the field.

3. The paper presents comprehensive experiments, including both objective metrics and human evaluations.

**Weaknesses:**

1. Novelty is moderate: While the system is well-engineered, many components, such as LLM-guided reasoning, semantic token bridging, and progressive fine-tuning, were builded upon established ideas in text-to-audio and multimodal generation (e.g., NExT-GPT, CoDi, TangoFlux). The LLM-DiT collaboration feels conceptually superficial, and the motivation for separating semantic and residual tokens seems ad hoc.

2. Dataset contribution is modest: the AS-10k benchmark is partly synthetic and generated using LLMs; its true diversity and annotation reliability are unclear. Additionally, I feel that the AS-10k dataset is only partially described. Data licensing, annotation quality, and source bias (especially with the use of animated clips from Tom & Jerry) are not discussed in depth, which raises reproducibility and ethical concerns.

3. Despite a “long-form” claim, most results appear under 2–3 minutes; it remains unclear whether the approach scales to truly long (e.g., 30+ minute) audio narratives without degeneration.

4. Missing comparisons to recent LLM-driven audio systems. Several state-of-the-art approaches: Make-an-Audio 2 , WavJourney, and ComposerX, also decompose narrative prompts via LLM reasoning into segmental descriptions before audio composition. These works must be discussed and compared, as their problem formulations and architectures (LLM decomposition + specialized audio modules) are conceptually similar.

Make-an-Audio 2:  https://arxiv.org/abs/2305.18474
ComposerX: https://arxiv.org/abs/2404.18081
WavJourney: https://arxiv.org/abs/2307.14335

**Questions:**

Please address my comments in weaknesses.

**Details Of Ethics Concerns:**

The AS-10k benchmark contains video clips from "Tom & Jerry".

---

### Official Review · Reviewer_jfsW · 2025-10-25

**Soundness:** 3
**Presentation:** 2
**Contribution:** 2
**Rating:** 2
**Confidence:** 5

**Summary:**

This paper presents a framework that integrates large language models (LLMs) with audio generation models for long-form narrative audio generation. The proposed system adopts a two-stage pipeline—first parsing and decomposing narrative structures, then generating corresponding audio events. Moreover, an end-to-end variant is introduced, enabling joint optimization of both the LLM and audio generator. The framework is multimodal, supporting text-, audio-, and video-based inputs. Additionally, the authors introduce a new benchmark dataset, AudioStory-10K, designed to evaluate long-form narrative audio generation.

**Strengths:**

•  The idea of combining LLMs and generative audio models for long-form narrative synthesis is novel and ambitious, particularly the multimodal compatibility (text, audio, and video).

•  The proposed semantic and residual tokens show some enhancement in controlling audio event composition, as demonstrated by ablation studies.

•  The introduction of a benchmark dataset (AudioStory-10K) provides a potential foundation for future work on long-form and story-driven audio generation.

**Weaknesses:**

•  The dataset heavily relies on LLM-based automatic annotation (e.g., Gemini, GPT), which raises concerns regarding accuracy and reliability. Further human validation is necessary to ensure the quality of captions, timestamps, and semantic annotations.

•  The end-to-end training strategy, including fine-tuning the LLM component, appears computationally expensive and conceptually unnecessary, especially since all training data are LLM-generated.

•  The paper lacks comparative analysis with existing strong baselines such as WavJourney[1], which already supports compositional and long-form audio generation without any complex retraining.

•  The proposed improvements on the audio generator side (semantic/residual tokens) seem to bring limited perceptual benefits, especially for single-event audio synthesis, is this module really necessary, and what’s the main difference on just directly using a well-trained state-of-the-art text-to-audio generator?

•  The overall results do not convincingly demonstrate a significant improvement over using existing LLM + audio generator combinations. The generated long-form audio sequences still sound disjointed, suggesting that cross-event coherence remains underexplored.

[1]. Liu, X., Zhu, Z., Liu, H., Yuan, Y., Huang, Q., Cui, M., Liang, J., Cao, Y., Kong, Q., Plumbley, M.D. and Wang, W., 2025. Wavjourney: Compositional audio creation with large language models. IEEE Transactions on Audio, Speech and Language Processing.

**Questions:**

1.	For the AudioStory-10K dataset, since most annotations are generated by Gemini or GPT, is there any human verification process to ensure the correctness of timestamps, captions, and event labels?

2.	Have you compared your framework’s performance—particularly in long-term narrative coherence or perceptual quality—against existing models like WavJourney?

3.	Could you clarify whether the end-to-end training brings measurable benefits compared to training the LLM and generator separately?

4.	Is there any plan to release human-evaluated subsets of the dataset to validate the reliability of LLM-based annotation?

---

### Official Review · Reviewer_DW3p · 2025-11-01

**Soundness:** 4
**Presentation:** 3
**Contribution:** 3
**Rating:** 6
**Confidence:** 3

**Summary:**

This paper proposes AudioStory, a system for generating long-form audio that is temporally aligned to an input source. To enable interleaved reasoning, the authors obtain detailed, timestamp-aligned sound descriptions from Gemini 2.5 Pro and then use GPT-4o to generate instructions and intermediate reasoning steps. Using these structured reasoning traces, the model performs divide-and-conquer generation by producing audio for each timestamped segment. Each segment is generated by a DiT-style flow matching model, and the paper proposes a bridging mechanism that maps LLM-derived representations into a form the DiT can effectively condition on, resulting in an end-to-end LLM-DiT model. Because training long-form audio generation directly is challenging, they adopt a progressive training strategy: starting from single audio clip generation -> unified generation and understanding → long-audio unified generation and understanding. The paper shows that the system achieves high consistency and strong instruction-following metrics for long-form audio generation, and provides ablation studies across multiple aspects.

**Strengths:**

- The demo samples follow instructions well and sound more natural and coherent. The demos also show that, compared to TangoFlux (a DiT initialized baseline), the proposed method produces significantly better audio, suggesting that the proposed training strategy is meaningful.
- The paper proposes a way to train an LLM and a DiT jointly end-to-end, and a progressive training strategy specifically designed for long-form audio generation. By training a reasoning-capable LLM together with a DiT, the system leverages interleaved reasoning to plan what should happen over long time spans and then realize that plan in audio, making long-horizon generation more consistent.

**Weaknesses:**

- The overall training setup is relatively complex, largely because it combines two very different model classes into a single system.

**Questions:**

- In the bridging-related ablations, could TangoFlux simply be benefiting from the fact that it was initialized with a text encoder (Flan-T5) that already produces features that are easier for the DiT to use, compared to other representations?
- If there were a DiT-based TTA-style model that directly consumes frozen LLM text representations, do you think end-to-end LLM-DiT training would still require a bridging mechanism (w/ $L_{bridge}$), or could we build a joint LLM–DiT model in a simpler way without bridging mechanism?
- In Figure 2, there is an “alignment loss,” but the text does not define it. Is that the same as $L_{bridge}$?

---

### Note · Authors · 2025-11-13

I have read and agree with the venue's withdrawal policy on behalf of myself and my co-authors.